# Ways into Understanding HIF Inhibition

**DOI:** 10.3390/cancers13010159

**Published:** 2021-01-05

**Authors:** Tina Schönberger, Joachim Fandrey, Katrin Prost-Fingerle

**Affiliations:** Institute of Physiology, University of Duisburg-Essen, 45147 Essen, Germany; tina.schoenberger@uni-due.de (T.S.); katrin.prost@uni-due.de (K.P.-F.)

**Keywords:** cancer, hypoxia, hypoxia-inducible factor, HIF, inhibitor, PHD, pVHL, FRET, visualization

## Abstract

**Simple Summary:**

Cancer cells adapt to hypoxia, survive, and grow. To that aim, they engage hypoxia-inducible pathways. These pathways are under intense investigation in search of new therapies to interfere with signaling components to kill cancer cells. Nowadays, new technologies enable more in-depth studies of hypoxia-induced signaling including protein–protein interaction and transcriptional processes, as well as the mode of action of different inhibitors. In this review, we give insight into useful techniques for studying the components of the hypoxia-inducible pathway and current inhibitors.

**Abstract:**

Hypoxia is a key characteristic of tumor tissue. Cancer cells adapt to low oxygen by activating hypoxia-inducible factors (HIFs), ensuring their survival and continued growth despite this hostile environment. Therefore, the inhibition of HIFs and their target genes is a promising and emerging field of cancer research. Several drug candidates target protein–protein interactions or transcription mechanisms of the HIF pathway in order to interfere with activation of this pathway, which is deregulated in a wide range of solid and liquid cancers. Although some inhibitors are already in clinical trials, open questions remain with respect to their modes of action. New imaging technologies using luminescent and fluorescent methods or nanobodies to complement widely used approaches such as chromatin immunoprecipitation may help to answer some of these questions. In this review, we aim to summarize current inhibitor classes targeting the HIF pathway and to provide an overview of in vitro and in vivo techniques that could improve the understanding of inhibitor mechanisms. Unravelling the distinct principles regarding how inhibitors work is an indispensable step for efficient clinical applications and safety of anticancer compounds.

## 1. Introduction

Hypoxia-inducible factors (HIFs) regulate a majority of oxygen-dependent genes and are considered to be the master regulators for cellular adaptation to low oxygen concentrations [1,2]. The dimeric transcription factor HIF is composed of one of three oxygen-regulated α-subunits (HIF-1α, HIF-2α, or HIF-3α) and a constitutively expressed β-subunit, named HIF-1β or ARNT (aryl hydrocarbon nuclear translocator) [3]. HIF-1α is found in all nucleated mammalian cells, while HIF-2α is characterized by a tissue-specific expression. HIF-1α/HIF-2α are highly homologous, with approximately 70% identity in their amino acid sequence and they function in a similar manner [4]. Much less is known about HIF-3α, which can be found in different splice variants and has been suggested to negatively regulate the transcriptional activity of HIF-1α and HIF-2α gene expression [5,6]. HIF-3α has recently been reported to be a target gene of HIF-1α and different HIF-3α isoforms, such as HIF-3α1 or HIF-3α9, were shown to influence the transcription of distinct hypoxia-related genes [7,8]. Herein, we focus on HIF-1α and HIF-2α as oxygen-dependent components of the HIF transcriptional complex.

## 2. Regulation of the Hypoxia-Inducible Factor (HIF) Pathway

Both HIF-1/2α subunits are regulated on the post-transcriptional level. Under normoxic conditions, two proline residues of the α-subunit are hydroxylated by prolyl-4-hydroxylase domain (PHD) proteins, and thereby marked for proteasomal degradation [9,10]. Since the activity of the three homologous hydroxylases PHD1, PHD2, and PHD3 is strictly dependent on oxygen concentration, they are considered to be the oxygen sensors of the cells [11]. An abundance of HIF-α and PHD isoforms is highly tissue-specific and driven by physiologically different pO_2_ values in distinct tissues [12]. Upon sufficient oxygen supply, oxygen coordinately binds to one Fe(II) atom in the active center of the PHD enzymes and initiates hydroxylation of HIF protein proline residues. This enables recognition by the von Hippel–Lindau tumor suppressor protein (pVHL) as part of an E3 ubiquitin ligase complex. Lysine residues surrounding these hydroxylated proline residues undergo ubiquitination, which finally labels HIF-α for proteasomal degradation [10,13,14]. Transcriptional activity of the HIF-α subunits depends on oxygen-dependent hydroxylation of asparagine residues in HIF-1/2α by the asparagyl hydroxylase factor-inhibiting HIF (FIH), preventing the binding of transcriptional coactivators (CBP/p300) and subsequent expression of target genes under oxygenated conditions (Figure 1) [9].

Under hypoxic conditions, the lack of oxygen reduces the enzymatic activity of PHDs and FIH and initiation of HIF-α degradation is blocked. The HIF-α subunits accumulate in the cytosol and translocate via a core localization sequence into the nucleus. Nuclear heterodimerization of the HIF-α subunit with the constitutively expressed HIF-1β subunit is mediated by the basic helix–loop–helix (bHLH) and Per-ARNT-Sim (PAS) domains of the proteins [15]. Subsequently, the N- and C-terminal transcription activation domains (N-TAD and C-TAD, respectively) enable transcription complex formation by recruitment of the cofactors CBP/p300 and Src-1 and binding to hypoxia responsive DNA elements (HRE) located in the promoter region of HIF target genes [16]. Thus, transcription by HIF is regulated by the abundance and binding affinity of the α subunits to regulate a variety of genes in physiology and pathophysiology [3,17]. The tumor microenvironment is known to essentially contribute to tumor progression, however, the distinct role of HIF is still under investigation [18]. Deregulation of the HIF pathway can occur on multiple levels in pathological settings. Mutation of the VHL gene, as found in clear cell renal carcinoma, leads to dysfunction of the VHL protein complex. This results in extensive accumulation of the HIF-α protein, even under normoxic conditions, which drives activation of tumorigenic target genes [19]. The inflammatory microenvironment in cancer is further characterized by a high content of reactive oxygen species (ROS). ROS increase the HIF-α protein stability by blocking PHD activity, and thereby protein degradation, independent of the oxygen concentration in the tissue [20,21,22] (Figure 1). HIFs are crucial to overcome potentially fatal hypoxic conditions in tissues, for example, during growth and inflammation, but they also allow malignant cells of multiple cancers to survive the hypoxic tumor environment and to grow outpacing the growth of normal cells. Then, tumor progression can be driven by hypoxia itself (Figure 1).

### Target Genes of HIF Transcription Factor

To counteract deadly hypoxia and to ensure cellular survival, cells need to restore their oxygenation by improving oxygen transport and supply on different levels. The most prominent example in this respect, which also leads to the identification of the HIF pathway and its role as the master regulator of hypoxia-induced gene expression, is the hypoxic production of erythropoietin (EPO) [1,13,23]. EPO elevates systemic blood oxygen capacity by increasing erythrocyte numbers, and therefore oxygen transport. Additionally, transcriptional upregulation of the vascular endothelial growth factor (VEGF) stimulates angiogenesis and ensures improved oxygen supply to affected regions by new capillaries. On the cellular level, adaption to ATP shortage is counter regulated via enhanced glucose uptake and glycolysis mediated by increased expression of glucose transporter 1 (GLUT1) [2,3,17]. Glycolytic enzymes such as phosphoglycerate kinase 1, lactate dehydrogenase-A, carbonic anhydrase 9, and aldolase are transcriptionally regulated by HIFs. Here, it is of particular interest that glycolytic enzymes are under the control of HIF-1α, while HIF-2α targets gene transcription of EPO, transforming growth factor alpha, and cyclin D. Nevertheless, there is redundancy in target gene regulation by HIF-1α and HIF-2α under specific conditions. Therefore Holmquist-Mengelbier et al. hypothesized that HIF-1α-dependent target gene transcription could be predominant in the acute hypoxic phase, while HIF-2α driven gene transcription could take over upon chronic hypoxia [24]. Important redundant target genes include VEGF, GLUT1, and adrenomedullin 1 (ADM-1) [25,26,27]. Several reviews have highlighted the differences of HIF-1α and HIF-2α regulation and target gene transcription [17,28,29].

Similar to other master regulators when deregulated, both HIF-1α and HIF-2α can drive pathologies, in particular tumorigenesis and cancer progression. Both are frequently found to be co-expressed in human cancers [25,30,31]. Especially in solid tumors, hypoxic regions develop when tumor cell growth outpaces vascularization; here, the ability to adapt to low oxygen conditions is crucial for survival and proliferation of cancer cells. A broad range of adaptive mechanisms induced by HIF-1/2α is linked to a more aggressive tumor type. The so-called angiogenic switch causes tumor angiogenesis, which often results in chaotic non-functional vessels and does not facilitate blood supply of the tumor; however, tumor cells may enter these capillaries and give rise to metastasis. The role of HIF-1/2α has been well studied for these processes and both isoforms have been identified as potential targets in antitumor therapy [32,33,34]. Recently, the role of HIF-3α has gained more attention from the field. HIF-3α was found to be overexpressed in pancreatic cancer, especially under hypoxic conditions promoting metastasis by activation of the ras homolog family member C/Rho associated coiled-coil containing protein kinase (RhoC-ROCK) pathway [35]. Tolonen et al. showed that HIF-3α directly regulates a subset of hypoxia-inducible genes involved in lipolysis (angiopoietin-like 4) and metabolism (angiopoietin-like 3 and pantothenate kinase 1) and, most interestingly, bound to the promotor region of the EPO gene. The authors suggested that this could indicate synergistic effects of HIF-1/2α and HIF-3α in terms of EPO transcription [36]. Overall, HIF-3 could have a dual role, making it interesting for further research in light of physiological regulation and its deregulation in tumorigenesis.

## 3. Targeting the HIF Pathway

Deregulation of the HIF pathway in cancers has been intensively addressed [3,28,31]. Hypoxia-related gene regulation is involved in enforcing tumorigenic, invasive, and metastatic potential and in accelerating resistance to chemo- and radio-therapy. Thus, to address the HIF regulatory pathway in therapeutic approaches is very attractive, but a detailed understanding of its components is essential for developing new drugs. In addition to HIF itself, other central components mediating the hypoxic response mechanism present potential drug targets, including the PHDs and pVHL or FIH and CBP/p300.

In recent years, several groups have focused on finding the Achilles’ heel, i.e., weakness, in the HIF pathway to target HIF activity in disease. It is obvious that this is notoriously complicated, since HIF, as a transcription factor, acts in the nucleus making it not easily accessible. Moreover, HIF is important for many physiological processes, thus, one has to be well aware of potential unwanted side effects.

In principle, attempts to modulate the HIF pathway in cancer are driven by the concept of inhibiting HIF, because HIF activity aggravates the disease. In other diseases such as chronic kidney disease with an underlying lack of EPO production, desired drugs can increase HIF activity, preferably by specifically targeting HIF-2α. This approach has been successful for inhibiting PHDs with prolyl hydroxylase inhibitors (PHI) and several drugs are already on the market in China or in advanced clinical trials [37,38,39].

### 3.1. Inhibitors of HIF-α Transcription, Translation, and Protein Stabilization

HIF inhibitors have quite a broad field of action. They target HIF on multiple levels, ranking from transcription and translation to the transcription of HIF target genes (Figure 2). The so-called cardenolides are steroids that transcriptionally inhibit HIF-1. One example is the compound SN38, the improved version of the metabolite CPT-11 (EZN-2208/PEG-SN38), which has shown effectiveness in preclinical models of glioblastoma and lymphoma [40]. The Ca^2+^ channel blocker NNC 55-0396 decreases mitochondrial ROS production, thereby blocks HIF-1 activation, and additionally increases HIF-1α protein hydroxylation and degradation and suppresses HIF-1α de novo synthesis [41].

The HIF-1α inhibitor PX-478 also acts on multiple levels. It decreases HIF-1α mRNA levels, blocks HIF-1α translation, and inhibits de-ubiquitination, which results in enhanced protein degradation [42]. A phase I dose escalation study in patients with solid tumors (NCT00522652) showed good tolerability [43]. Recent studies have reported efficiency against pancreatic ductal adenocarcinoma in combination with chemotherapy (or combination therapy with gemcitabine) and inhibition of esophageal squamous cell tumor growth and immune modulation in mice [44,45].

Similarly, the well-known proteasome inhibitor bortezomib (PS-341/Velcade^®^, Millenium Pharmaceuticals, Inc., Cambridge, MA, USA) has been shown to repress HIF-1α on transcriptional and translational levels, as well as to inhibit the recruitment of the coactivator p300, blocking the PI3K/Akt/TOR and MAPK pathway (Figure 2) [46,47]. In 2003, bortezomib was approved by FDA for the treatment of multiple myeloma [48]. In addition, bortezomib as a proteasome inhibitor has increased HIF-1α protein level in monocytic leukemia cells [49].

Most current inhibitors focus on the interference with synthesis and stabilization of HIF-α protein, activation of PHDs, or binding of HIF-αs to pVHL. Probably due to its ubiquitous expression, inhibitors for HIF-1α protein have been more intensively investigated than those for HIF-2α. One approach, among several substances under examination, is to directly accelerate HIF-1α protein degradation (Figure 2). This can be achieved via upregulation of pVHL expression by the arytoxy acetylamino benzoic acid analogue IDF-1174 which leads to HIF-1 protein degradation [50]. Likewise, panobinostat (LBH589), developed for solid and hematologic cancers, acts as a histone deacetylase (HDAC) inhibitor and disrupts the Hsp90/HDAC6 complex [51]. Since Hsp90 complexing with HIF-1α, and also acetylation of HIF-1α, have been claimed to prevent degradation through the proteasome/pVHL pathway complex, panobinostat reduces HIF-1α protein. In a similar mode of action the indole-3-ethylsulfamoylphenylacrylamide compound MPT0G157 acts via inhibition of multiple histone deacetylases (1, 2, 3, and 6) and decreased HIF-1α protein in colorectal cancer [52]. Alternatively, HIF-1α protein is reduced by the use of diazepinquinazolin-amine derivate BIX01294 which increases PHD2 and pVHL expression [53] or the drug benzopyranyl 1,2,3-triazole inducing HIF-1α hydroxylation and ubiquitination, leading to increased protein degradation [54]. Finally, kresoxim-methyl analogues have been shown to promote proteasomal degradation of HIF-1α via increased oxygen tension in cancer cells [55].

A different approach is to address the accumulation of HIF-1α protein using nanoparticles such as CRLX-101 that suppress HIF-1α protein translation and stability [56] or the substance class of so-called glyceollins from the soybean which block HIF-1α translation via inhibition of the Pi3K/AKT/mTOR pathway and decrease HIF-1α stability by decreasing Hsp90 binding (Figure 2) [57].

Tumorigenesis correlates to irregular histone modifications and genetic instability [18,58,59,60]. HDAC enzymes physiologically tighten the chromatin structure, thereby, repressing gene expression. In addition, HDACs are also known to target non-histone proteins such as heat shock protein 90 (Hsp90). Any defect in HDAC function could, therefore, affect the chromatin structure and genetic stability. To address effects of deregulated HDAC in cancer, HDAC inhibitors are currently under investigation or are already in clinical use as anticancer agents. Vorinostat [46], a HDAC pan inhibitor, was approved for the treatment of cutaneous T cell lymphoma by the FDA, in 2006 [61]. Vorinostat in vitro affected multiple HDACs in glioblastoma, osteosarcoma, and hepatocellular carcinoma cell lines and inhibited hypoxia signaling by lowering HIF-1α and VEGF levels. The distinct mode of action is still not fully understood. The hypotheses rank from direct HIF-1/2 acetylation and degradation by pVHL to interactions with the HsP70/90 chaperone axis and decreased HIF-1/2 nuclear translocation [62]. Other promising and already approved HDAC inhibitors are romidepsin (FK228) [63], belinostat (PXD-101) [64], panobinostat (LBH-589) [65], and chidamide [66], for which effects on HIF have also been described by E. Pojani and D. Barlocco (Figure 2) [67].

HDAC inhibitors seem to be further qualified as a treatment option for neurological diseases characterized by oxidative stress-related cell loss [68], including multiple sclerosis [69], Parkinson’s disease [70], and stroke [71]. HDAC inhibitors act by acetylation of histones and, moreover, via modulation of numerous proteins, such as transcription factors. More specifically, pan-HDAC inhibitors have been shown to protect from oxidative stress-induced cell death (ferroptosis) by increasing DNA binding of the HDAC1-associated transcription factor Sp1 [72]. In line with that, inhibitors against the HIF prolyl hydroxylases could be valuable for fighting neurological diseases, due to the function of PHDs as epigenetic iron sensors, possibly driving oxidative stress-induced cell death, and thereby neurological disease progression [73]. This marks these inhibitor classes as important possible drugs utilizable for therapy of diseases beyond cancer.

Nevertheless, in cancer therapy, one focus could be to target the HIF system using compounds that promote PHD activity. Such drugs might function by strengthening the protein interaction between PHDs and HIF-α or enhancing the enzymatic activity of PHDs by increasing oxygen availability at the enzymatic site. This could also be achieved by pharmacological inhibition of the mitochondrial respiration chain, reducing its oxygen consumption and redirecting oxygen from mitochondria to PHDs [74]. Much less work, so far, has been concentrated on the activation or increased activity of the 2-oxoglutarate dependent oxygenase FIH. One may argue that, as compared with the PHDs, a fundamental role of FIH for the regulation of the hypoxic response has not been found for many HIF-dependent genes and, similarly, with FIH inhibitors regarding their efficacy for boosting the HIF response [75]. In principle, redirecting oxygen from mitochondria to FIH (as described for PHDs above) should increase FIH even more than PHD activity, due to the higher oxygen affinity of FIH as compared with PHDs [76].

### 3.2. Inhibitors of HIF-α/β Dimerization and Transcription Complex Formation

Essential for the activation of the HIF complex as a transcription factor to induce the expression of HIF target genes is the preceding dimerization of the HIF-α and HIF-β subunit. Obviously, interfering with the dimerization process is an attractive target for inhibitors of HIF because, in contrast to the above-mentioned drugs, such an approach could be highly selective and HIF specific. Thus, dimerization marks a promising target for pharmaceutical interference (Figure 2).

In 2009, Scheuermann et al. identified a cavity within the HIF-2α PAS-B domain, bearing room for small molecules and also, potentially, inhibitors that could interfere with the heterodimerization of the HIF-2 protein complex [77]. Closer investigation of this newly found binding pocket resulted in the development of HIF-2α inhibitors PT2385, PT2399, and later PT2977 [78,79,80]. These inhibitors from the same substance class have been or are currently under investigation in phase II clinical trials targeting clear cell renal cell carcinoma (PT2385: NCT03108066), recurrent glioblastoma (PT2985: NCT03216499), and VHL-associated RCC (PT2977: NCT03401788). A second-generation inhibitor PT2977 was generated by the exchange of a geminal difluoro group with a cis-vicinal difluoro group and it has raised great expectations with respect to efficacy. Another potent specific HIF-2α inhibitor is the compound 0X3, which also binds to the PAS-B domain and possibly alters the dynamics of the domain. Co- and chromatin-immunoprecipitation (ChIP) experiments have demonstrated disruption of HIF heterodimer formation and prevention of HRE binding [81]. Although molecular dynamics simulations revealed an interruption of crucial hydrophobic interactions within the HIF-2 dimer for the inhibitors PT2399 and 0X3, this has to be further verified in vivo [82]. State-of-the-art live cell microscopy could further elucidate mechanisms of action. Unexpectedly, however, first resistance mechanisms have been observed, highlighting the need for a better understanding of the mode of action of these inhibitors [83,84].

In a similar manner, the compound acriflavine (ACF) directly binds to the PAS-B domain of HIF-1α and HIF-2α with nanomolar affinities and blocks heterodimerization with HIF-1β [85]. The FDA-approved drug was studied by using a cell-based luciferase assay, which consisted of a split version of the *Renilla* luciferase partly coupled to HIF-α and HIF-β fusion proteins. Measurable bioluminescence was only emitted upon dimerization. This method made it possible to scan 200 currently in use drugs for their potential to inhibit HIF dimerization in cells and resulted in the identification of acriflavine that was reported to decrease tumor growth and vascularization [85]. Similar to ACF, the peptide cyclo-CLLFVY was investigated by luciferase reporter plasmids and specifically inhibited HIF-1 dimerization (Figure 2) [86].

Different cancers are associated with mutations or overexpression of HIF coactivators. In these cases, obstruction of transcriptional coactivators CBP and p300 appears to be an attractive strategy to inhibit the HIF pathway. Two domains, the histone acetyltransferase [47] and the bromodomain (BRD), are found in both coactivator proteins (CBP/p300) and present promising targets for pharmacological intervention [87,88]. For further reading about BRD inhibitors classified by their chemotypes, we refer to a recent review [89]. Currently, there are two clinical trials running (phase I/IIa studies) that are investigating the bromodomain inhibitor CCS1477. One trial (NCT03568656) addresses the potential of CCS1477 against advanced solid tumors and metastatic prostate cancer. The other trial focusses on treatment of patients with advanced hematological malignancies [90]. Another BRD inhibitor (CG1350) suppressed multiple myeloma cell proliferation in human cell lines and mice [91]. These (BRD) inhibitors have in common that they target protein–protein or protein–chromatin interactions. Thus, when screening for new drug compounds, fluorescence resonance energy transfer (FRET)-based methods are perfectly suited because they can demonstrate protein–protein interaction (see Section 3). In particular, recently reported time-resolved FRET-based high-throughput assays can be important tools to discover new potent drugs [92]. The BRD inhibitor (CG1350) is an example of a successful application of such methods by taking advantage of chromatin immunoprecipitation and a coupled luminescent-fluorescent method proving the advantages of techniques to study inhibitor binding and protein interaction in vitro. Nevertheless, more profound methods addressing binding dynamics and structural changes in vivo will help to further characterize these compounds.

### 3.3. Inhibitors of HIF Targets

A recent review from the Nobel prize laureate William G. Kaelin Jr. and his colleague Toni K. Choueiri provided a comprehensive overview of the biological background and clinical development of HIF-2α inhibitors to treat renal cell carcinoma [19]. The majority of clear cell renal carcinoma (ccRCC) tumors are characterized by mutations of the VHL gene, encoding for VHL protein. As described above, pVHL is part of an E3 ubiquitin ligase which recognizes hydroxylated HIF-α proteins, poly-ubiquitinates them, and marks them it for proteasomal degradation under normoxic conditions [10]. With dysfunctional VHL in ccRCC, HIF-α subunits accumulate independent of hypoxia and drive enhanced tumorigenic HIF target genes such as VEGF-A, which is, among epithelial cancers, the highest expressed in ccRCC [93]. Inhibitors against VEGF, including bevacizumab, sunitinib, and sorafenib, rarely trigger complete responses [94,95,96] and most combination therapies fail to improve patient outcome. Nowadays, immune-checkpoint blockers (ICBs) are the main strategy to fight kidney cancer, in addition to the in-use inhibitors against VEGF and the mTOR pathway blockers such as rapamycin analogues, everolimus, and temsirolimus [97]. To improve treatment for metastatic ccRCC, substances are combined in the treatment regime, however, it would be much better to develop new drugs.

## 4. Toolbox to Visualize HIF and Its Inhibition

It is fundamental to understand the pharmacological mechanism underlying small molecule inhibitors and their binding to targets for identification of their therapeutic potential, optimal application, and consideration of side effects. Large-scale biochemical analysis of inhibitors often fails to mimic the target engagement in living cells, leaving a gap for more conclusive methods. Qualitative methods to study compound binding are mostly based on ligand-induced protein stabilization with limited stability or need for non-physiological temperatures [98]. Hence, the use of in vivo methods, such as fluorescence resonance energy transfer (FRET) or bioluminescence resonance energy transfer (BRET), can be an asset to determine target engagement in real time, in living cells and tissues.

The use of many HIF inhibitors is dampened by their off-target effects, affecting different pathways in DNA replication, cell division, or cell signaling. Hence, it is important to unveil the distinct mechanism of action to provide efficient and safe compounds for cancer treatment.

A combination of different microscopy methods, such as FRET [99] or fluorescence lifetime imaging microscopy (FLIM) was a significant step forward in understanding protein–protein interactions and HIF complex formation in living cells. In addition to the knowledge gained from conventional protein-biochemical approaches, live cell imaging interaction studies have provided insight into the spatiotemporal regulation of HIF interaction. During the last years, a combination of multiple methods has paved the way for establishing anticancer drugs that target HIF protein dimerization, and thus prevent the activation of deregulated HIF target genes.

FRET is a non-radiative energy transfer between a donor and an acceptor fluorophore. When in close proximity (< 10 nm), emission energy of the donor may non-radiatively transfer in a dipole–dipole reaction, and thereby excite the acceptor molecule [100]. The emission spectrum of the donor needs to overlap with the excitation spectrum of the acceptor to allow the dipole–dipole interaction of both molecules (Figure 3). To study protein–protein interactions and protein complex formation, labeling of the protein regions of interest with corresponding monomeric fluorophores provides an elegant tool for investigations [101,102,103], since the FRET signal between fluorophores fused to interaction partners correlates strongly with physical interaction. In combination with other fluorescence approaches, this method enables greater insight in the field of hypoxia research and the distinct mechanism of dimerization of the HIF subunits (HIF-1/2/3 α and HIF-1β).

Immunofluorescence combined with two-photon microscopy unveiled the subnuclear distribution of HIF-1α and interacting factors such as HIF-1β and it has shown that proteins colocalize in the nucleus in speckle-like structures under hypoxic conditions [104]. Additional work using fluorescence recovery after photobleaching (FRAP) have reported faster mobility of HIF-1β as compared with HIF-1α, in cells cotransfected with both fluorescently labeled proteins [105]. Using FRET in combination with enhanced yellow fluorescent protein (EYFP)- or enhanced cyan fluorescent protein (ECFP)-labeled HIF proteins, further analysis of the HIF complex formation in the nucleus in vitro showed a heterogeneous subnuclear distribution of the HIF-1 α/β heterodimers. Moreover, the FRET approach proved the compact assembly of the HIF-1 complex (6.2 to 7.4 nm between the N-and C-termini of the HIF-1 subunits) [106]. Analogous to the interaction of the HIF-1 α/β, the HIF-2 α/HIF1-β heterodimers were proven in living cells with their typical nuclear localization [41,107]. Beyond that, quantification of interacting HIF fractions and potential reduction upon inhibitor application of such interaction has been analyzed using FLIM-FRET microscopy [108]. Detailed analysis of HIF protein regulation and complex formation has enabled the identification of binding sites within the HIF protein, which turned out to be druggable targets within the HIF pathway [77,81]. Testing such drug candidates in live cell FRET microscopy offers a direct observation of their effects, for example, disruption of protein-DNA binding, preventing initial dimerization, and enhancing protein degradation.

Further validation of in vitro results requires studying the HIF pathway in vivo, in appropriate animal models. Near-infrared (NIR) in vivo light imaging has been successfully established [109], although bioluminescent imaging methods with a better signal-to-noise ratio can be used independently from external excitation sources. Fluorescent and bioluminescent proteins can be genetically modified so that close proximity to specific protein areas or enzymes influence their activity or abundance. In these systems, the energy of a bioluminescent donor is transferred to a fluorescent acceptor molecule when in proximity. Similar to in FRET, the spectral overlap of the spectrum of donor emission and that of an acceptor excitation is a prerequisite. Multiple variations of this approach have been developed and have enlightened the understanding the role of hypoxia in disease models. In one example, fusion of gene sequences for the HIF-α oxygen dependent degradation domain (ODD) which causes oxygen lability of HIF-α subunits to luciferase sequences (ODD-Luc) was used to engineer a transgenic reporter mouse. These mice accumulated luciferase protein in all their hypoxic tissues where a lack of PHD dependent hydroxylation of the ODD prevented degradation of the ODD luciferase fusion protein. Imaging the development of tissue hypoxia and also the effect of prolyl hydroxylase inhibitors (PDI) was possible in the mice [110].

The use of genetic bioluminescence reporter assays and bioluminescence resonance energy transfer has further improved the understanding of protein interactions and DNA binding events in vivo. P. Iglesias and J. Costoya used an innovative hypoxia-biosensing system by combining a tracer module with near-infrared fluorescence and bioluminescence (mCherry-luciferase fusion protein), which was activated by HIF-1α. This tool was used to visualize HIF-1α protein accumulation in vivo, in mice with xenograft tumor implants. Obviously, such imaging systems improve our understanding of HIF-1 protein activation and, in particular, the development and role of biological relevant hypoxia in solid tumor development and tumor progression [111]. The detection of HIF-1 activation in vivo is superior to any other hypoxia staining methods or even physical measurements of oxygen tension, since HIF-1 activation reflects the tissue response to low oxygen tension. Other studies have proven the advantages of this system by using injectable imaging probes with a single donor and acceptor (NIR-BRET) for non-invasive detection of the activity of factors regulated by the ubiquitin-proteasome system, such as the HIF-α subunits, in different in vivo cancer models [112]. Among studies in solid tumors, innovations such as self-luminescing BRET-FRET NIR-emitting nanoparticles have enabled in vivo mapping of lymphatic networks and small metastases [113]. Especially for studying difficult-to-access pathologies such as age-related macular degeneration [114] and diabetic retinopathy, biosensors can bring a substantial advantage for treatment decisions. Often, these pathologies are characterized by an inflammatory hypoxic environment that results in the upregulation of the HIF target VEGF and pathological neovascularization in the eye [115]. BRET VEGF biosensors, using *Renilla* luciferase and extracellular IgG-like domains binding to VEGF, have been shown to quantify VEGF expression in vitro, leading the path to in vivo VEGF measurements in a patient’s eye [116].

In addition to the investigation of hypoxia and related mechanisms, these techniques have also been used to profile binding kinetics and target engagement for inhibitors in living cells. In 2015, M. Robers and colleagues demonstrated a method to quantitatively assess target engagement of the HDAC inhibitor romidepsin (Istodax/FK228) by using BRET in genetically modified cells expressing an intracellular target protein fused to luciferase and a fluorescent compound tracer [117].

Chromatin immunoprecipitation (ChIP) has been widely used to study protein-DNA interactions, and thereby transcriptional initiation of directly activated transcription sites of target genes. Specific proteins, including histones, transcription factors or cofactors, or modifications of these proteins chemically crosslinked to their cognate DNA sequence can be used as targets for precipitation by antibodies, resulting in immune-enriched DNA fragments. These can be subsequently quantified by qPCR or next generation sequencing (NGS) to investigate protein binding to a genomic region/target gene (Figure 3) [118]. Further distinctions are made between ChIP with crosslinking of proteins to proteins and proteins to DNA (X-ChIP) and without crosslinking, named native (N)-ChIP. Whereas N-ChIP is characterized by a predictable antibody specificity and efficient precipitation, X-ChIP has the advantage of being applicable to non-histone proteins with weak binding affinities to DNA with minimal histone rearrangements [119,120]. ChIP can provide valuable information about the binding of nuclear proteins to specific DNA sites, as well as posttranslational modifications of nuclear proteins [121]. However, sometimes the lack of specific and suitable antibodies and quantification of big protein complexes limit this method.

A relatively new approach in cancer theragnostics is the application of so-called nanobodies (Nb) or nanobody-based delivery systems. Nanobodies are, broadly speaking, the variable domain of engineered heavy chain (only) antibodies (HCAbs) in camelidae. Recombinant production of the heavy variable domain (VHH) of the antibody results in the generation of a single domain antibody (sdAB), also known as nanobody (Nb) with full antigen-binding potential [122]. Nanobodies are characterized by high immune compatibility, good solubility, robustness against unsteady pH and temperature values, and thanks to their small size, high infiltration rates into tissues and small cell interfaces or into cells [123,124]. Nanobodies can be further engineered by fusion with another monovalent nanobody, recognizing different epitopes on the same molecule or even binding to different target molecules (Figure 3). In addition, coupling to other molecules (such as albumin for a half-life increase) or conjugation with various drug candidates has displayed wide ranging applicability. For further reading we refer to recent reviews [125,126].

Furthermore, nanobodies serve as an elegant tool for in vitro and in vivo visualization in disease models, as well as in cancer diagnosis and monitoring [127]. To that aim, nanobodies specific for epitopes such as CD20 [128], epidermal growth factor 3 [129], or dipeptidyl-peptidase 6 [130] are fused with the enzyme sortase [131], to create an antigen specific non-invasive imaging probe (Figure 3). During the last years, the spectrum of methods have become divers, including GFP-binding nanobodies coupled to a DNA oligonucleotide-wrapped single-walled carbon nanotube (SWCNT) emitting NIR fluorescence to monitor neurotransmitter and motor proteins in living Drosophila melanogaster embryos [132]. Final imaging is mostly performed with positive electron tomography computed tomography (PET-CT) or single photon emission computed tomography (SPEP-CT). Imaging of tumor and inflammatory tissues with nanobodies takes advantage of the fast penetration of the probes into the target tissue, low background noise, and the possibility to trace distribution in tumors and of invaded immune cells in vivo, which makes it a promising method for future studies.

## 5. Conclusions

Versatile imaging methods have been combined with classic protein biochemical analysis to enable accurate investigation of HIF protein accumulation, HIF protein–protein interactions, and HIF transcription factor complex assembly. Moreover, the molecular mechanisms underlying the inhibition of these processes can be assessed and optimized. Many compounds target the HIF pathway on multiple levels, making it difficult to differentiate side effects of these compounds from their HIF specific effects. For the safety and optimal usage of inhibitory compounds, it is indispensable to unveil their distinct modes of action by in vitro and in vivo methods. While all inhibitors should finally target HIF-dependent gene expression, they are grouped into classes addressing different levels in the HIF pathway, from gene expression to protein synthesis and protein stabilization to dimerization of the subunits. The imaging methods with luminescence and fluorescence approaches, such as FRET and BRET, in living cells may improve spatiotemporal profiling of the mode of action, unravelling unexpected effects and helping to improve the specificity and efficacy of HIF inhibitors.

## Figures and Tables

**Figure 1 cancers-13-00159-f001:**
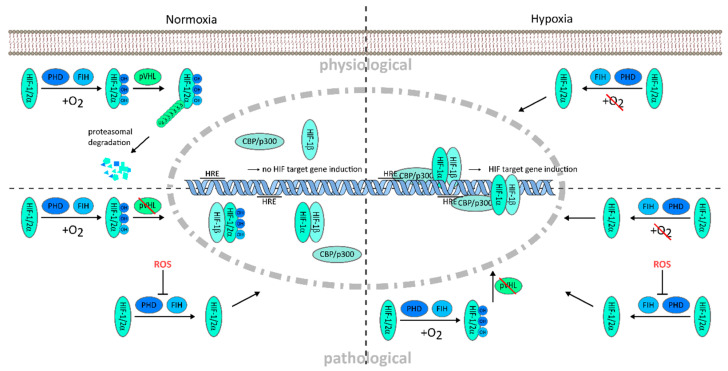
Oxygen dependent hypoxia-inducible factor (HIF) regulation and deregulation of HIF in tumor environment. Normoxic oxygen conditions lead to proteasomal degradation of the hypoxia-inducible factor (HIF) protein and prevent target gene transcription (**upper left**); Decreased oxygen availability deactivates the hydroxylation of the HIF protein by cofactors, factor-inhibiting HIF (FIH) and prolyl hydroxylases (PHD), which enables HIF protein accumulation and translocation into the nucleus. HIF-α and -β subunits dimerize and form a transcription complex with the cofactors CBP/p300. Complex binding to the hypoxia responsive elements (HRE) on the promoter region of HIF target genes leads to their transcription (**upper right**); Under pathological conditions, hypoxia can be accompanied by increased reactive oxygen species (ROS) and impaired von Hippel–Lindau protein (pVHL) function, promoting HIF induction to an excessive extent, e.g., driving tumorigenesis under normoxic, as well as hypoxic condition (**lower left and right**). OH, hydroxylation; Ub, ubiquitination.

**Figure 2 cancers-13-00159-f002:**
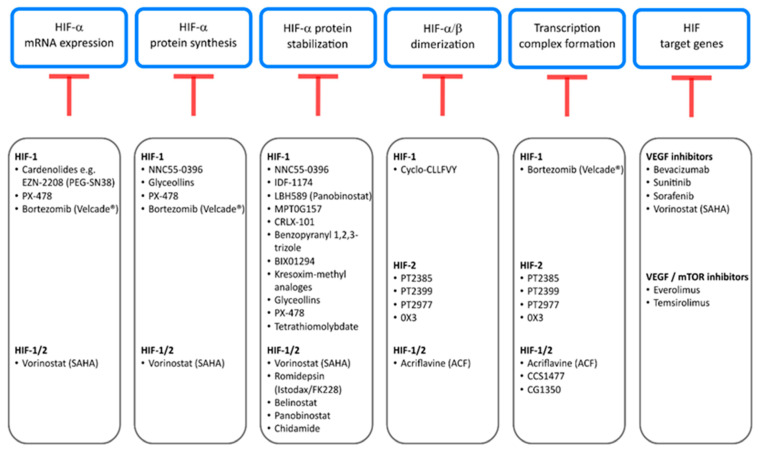
Inhibitors of the HIF pathway and intervention nodes summarized in this review. HIF inhibitors are broadly classed into their mode of action, targeting different levels of the HIF pathway. Starting from blockage of HIF mRNA expression to, finally, HIF target gene transcription and inhibition of the downstream genes, such as VEGF itself. For detailed information see Section 2.

**Figure 3 cancers-13-00159-f003:**
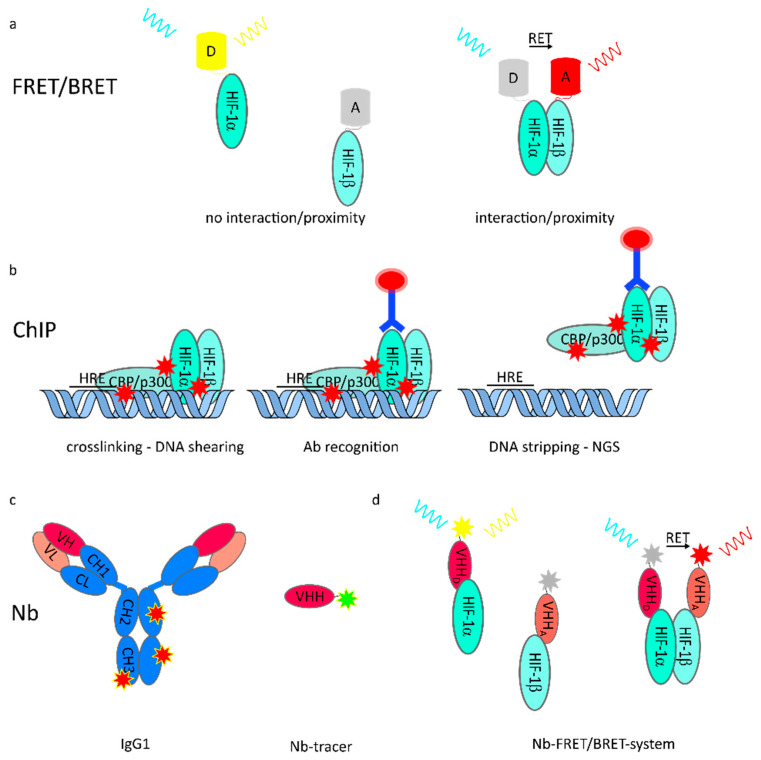
Methods to unravel effects on HIF protein stability, interaction, and function in vitro and in vivo. (**a**) Interaction of protein (complexes) is detectable by proximity-depended resonance energy transfer such as fluorescence resonance energy transfer (FRET) or bioluminescence resonance energy transfer (BRET). Protein localization and proximity are directly imaged in vitro or in vivo and recorded with high spatiotemporal resolution; (**b**) Transcriptional activity of transcription factor is assessed by chromatin immunoprecipitation (ChIP). Precise antibody recognition and accuracy is mandatory for qualitative evaluation of changes upon any perturbation; (**c**) Nanobodies (Nb), shown in relation to conventionally used IgG1, can be functionalized and used as intravital tracers. Due to their small size, penetration depth of cells and tissue is superior. Stoichiometric labeling of Nb enables quantitative imaging; (**d**) Nb tracer addressing interaction partners allow in vivo FRET/BRET independent of genetic manipulation, giving rise to a multitude of new life cell/intravital interaction studies.

## Data Availability

Data sharing not applicable. No new data were created or analyzed in this study. Data sharing is not applicable to this article.

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
