# Peer review of "Ways into Understanding HIF Inhibition"

_cancers, 2021, doi:10.3390/cancers13010159_

Round 1

Reviewer 1 Report

This is a well written review that summarizes the existing knowledge of the current inhibitor classes targeting the HIF pathway and the vivo techniques that could improve the understanding of the inhibition mechanisms.

I suggest that the authors explain in more detail the various aspects of the deregulation of HIF in the tumour microenvironment presented in Figure 1, and merge the paragraphs “Inhibitors of HIF-a mRNA expression and translation” and “Inhibitors of HIF- protein synthesis and stabilization”.

Author Response

Thank you for the fast response and your suggestions for improvement of our review “Ways into understanding HIF inhibition”.

We thankfully implemented your suggestions and look forward to your feedback.

Following adaptions to the manuscript were made:

According to your suggestions:

  • Addition to section 1, “Regulation of the HIF pathway”:
    Explanations of the deregulation of the HIF pathway in the tumour microenvironment with connection to figure 1, page 3 were added (line 82 - 91).
  • Merge of the paragraphs “Inhibitors of HIF-a mRNA expression and translation” and “Inhibitors of HIF- protein synthesis and stabilization” in section 2 “Targeting the HIF pathway”, page 4-5 to “Inhibitors of HIF-α transcription, translation and protein stabilization” (line 156, 175).

Reviewer 2 Report

  1. The authors should provide some description of methodology of how the articles included in present review were identified. For example, which database was searched, and what search terms used? Were references in selected articles also included for consideration? Which articles were excluded? For example time period, letter and review types?
  2. The authors should provide the information whether and to what extend was the PRISMA statement followed.
  3. Page 3, the phrase “Tolonen et al. showed that HIF-3α is directly regulates…” is grammatically incorrect. Please correct.

Author Response

Thank you for the fast response and your suggestions for improvement of our review “Ways into understanding HIF inhibition”.

We thankfully implemented your suggestions and look forward to your feedback.

Following adaptions to the manuscript were made:

According to your suggestions:

  • Correction of phrase “Tolonen et al. showed that HIF-3α is directly regulates…” into “Tolonen et al. showed that HIF-3α directly regulates…” (page 3, section “Target genes of HIF transcription factors”, line 128)

  • Description of methodology and review structuring according to the PRISMA statement:

The literature was mainly selected online using the database PubMed (US National Library of Medicine National Institutes of Health), in addition the “Google scholar” function by Google was used to access basic works. Literature search was focused on the according review section. Following search terms were used solely or in combination: HIF pathway, hypoxia-inducible factor, HIF(-α/β), hypoxia, cancer, tumour, HIF dimerization, inhibitor, Histone deacetylase, HIF antagonist, visualization, fluorescence, luminescence, imaging, CHIP, nanobody.

After detailed analysis of the literature, reference articles of the selected articles were also included to provide a more in-depth understanding of the regarding topic and the primary sources.  In addition, emphasis was placed on international and frequently cited literature (research articles) from the last 10 years. To complement our review, we referred to recent reviews from the field.

The review was structured according to the PRISMA Statement and the Checklist of 2009 (http://www.prisma-statement.org/PRISMAStatement/Checklist.aspx). Whenever possible we followed the statement. Sections regarding methods (e.g. study selection, data items, summary measures) and results (e.g. synthesis of results, additional analysis) were excluded when they were not applicable to the review format.